# Continual Causal Abstractions

**Matej Zečević**[†]     **Moritz Willig**     **Jonas Seng**     **Florian P. Busch**[+]

Computer Science Department, AIML, TU Darmstadt, Germany
[+]Hessian Center for AI (hessian.AI), Germany
[†]correspondence: `matej.zecevic@tu-darmstadt.de`

## Abstract

This short paper discusses continually updated causal abstractions as a potential direction of future research. The key idea is to revise the existing level of causal abstraction to a different level of detail that is both consistent with the history of observed data and more effective in solving a given task.

**Overview.**  [1] discusses the necessity of (a) causal abstractions for effectively solving tasks and (b) continual updates when data starts changing. [2] highlights existing approaches for (a). [3] discusses starting points for (b).

**Abbreviations.**  Structural Causal Model (SCM), Continual Causal Abstractions (CCA), Cognitive Science (CogSci), Artificial Intelligence (AI).

## [1] Motivation

Both causality (as defined by Pearl (2009)) and continual learning (Hadsell et al. 2020) seem integral for the quest of understanding (artificial) intelligence. Causality's main contribution has been the *formalization* of key ideas such as interventions, counterfactuals and structural mechanisms. Continual learning on the other hand raised awareness for the importance of learning stable concepts *continuously over time* void of access to previous experiences much like biological systems.

Many different research directions have been explored in the realm of both areas to (a) boost the performance and applicability of methods (Kyono, Zhang, and van der Schaar 2020; Zečević et al. 2021; Nilforoshan et al. 2022; Mundt et al. 2022b) but also to (b) understand the challenges that lie ahead of these and future methods (Schölkopf et al. 2021; Zečević, Dhami, and Kersting 2022; Mundt et al. 2022a).

The goal of the AAAI 2023 Bridge Program on "Continual Causality" lies in finding answers to the question of what may be found at the interesection of the two subfields primarily studied in AI and CogSci research. This short paper envisions the extension of existing work on causal abstractions towards a continual learning setting where the task solving agent ought to revise its current model abstraction towards a new level of detail in order to be consistent with the history of observed data and also find more effective decision rules for solving the current task.

Published at AAAI 2023 Bridge on Continual Causality.

**Why Do We Need Causal Abstractions?**  Let's illustrate with a simple example. Imagine being a chemist analyzing a particular gas. Your advisor tasks you to analyze the temperature and pressure in the volume. Using your thermo- and barometer you simply measure the desired quantities. The first task is solved. Next, your advisor tasks you to analyze the average velocity of a moving particle within the gas. It quickly becomes apparent to you that the previous model of the gas becomes obsolete, since the question has shifted from a macro- to a microscopic level of detail in which we suddenly need information on individual particles. Put differently, we are in need of a different level of *abstraction*. Not just that, we further want our new model to still be consistent with our previous observations, that is, if we were to measure net kinetic energy of all relevant particle combinations, then we would ideally like to see a match with the previous measurements of the thermometer. This requirement is what is being captured by the *causal* part of the abstraction transformation between the two models for each of the tasks.

**Why Do We Need Continual Abstractions?**  Let's illustrate again using an example. The following example is a sneak peek into the example from the vision schematic presented in Fig.1. Imagine being a dietitian analyzing the (causal) effect of a particular diet onto the risk of heart disease. Your history of clients has taught you that the total cholesterol of the patient is characteristic of whether or not the risk of heart disease for that individual is increased or not. This is your initial, base abstraction: if the diet is balanced, the cholesterol levels will deteriorate and the risk of heart disease decreases. Now a new client, a sumo practioner, enters your diet program but ends up overdoing it and eating three times the amount of items listed in the plan. To your surpise, although the cholesterol levels of the sumo were increased, his risk of heart disease had decreased. To cope with this new counterexample to the previous hypothesis, you decide to revise your abstraction as you found the high- and low-density lipoproteins to be more predictive of the risk of heart disease. For the sumo's case, it was that the latter increased the total cholesterol levels while still lowering the risk of heart disease. In other words, the dietitian *continually* updated the current best causal abstraction to comply with the data history while still answering the initial scientific question effectively.

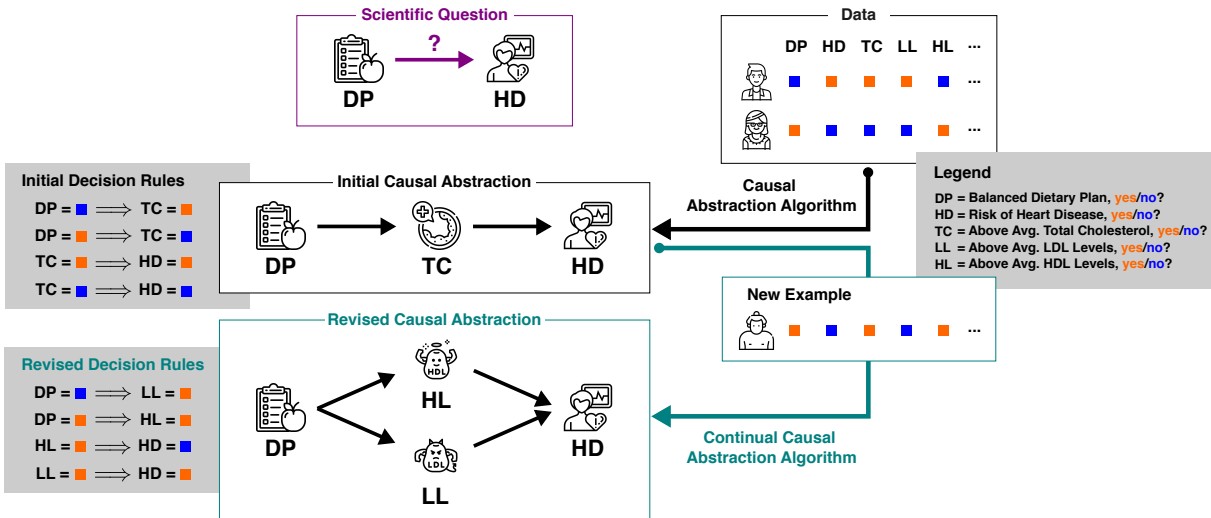

Figure 1: **Vision of Continual Causal Abstractions.** A schematic illustrating how CCA could work. The scientific question (top purple box) asks about the causal relation (or effect) of a balanced diet onto the risk of heart disease. Based on data consisting of patient records recording various features (see right grey box for a legend), a causal abstraction algorithm provides the initial causal abstraction (middle box) that suggests optimal decision rules (middle left grey box) based on the mediator variable of total cholesterol. With the incoming new example (right teal box) the CCA algorithm provides an updated causal abstraction that uses both HL and LL as mediator variables (middle teal box) with new optimal decision rules (lower left grey box). In summary, the macroscopic view of cholesterol levels in the initial abstraction was sufficient for analyzing the initial two data points, however, the third data point required a more fine grained abstraction that considers the levels of high- and low-density lipoproteins since an increase in former also leads to an increase of TC but is actually lowering HD. (Best viewed in color.)

## [2] Existing Work on Causal Abstractions

**Definitions.** The study of causal abstractions is a subfield in Pearlian causality that aims at *formalizing* the philosophical concept of an abstraction such that the resulting definition is maximally "useful in practice" (commonly taken to mean that the examples such as that of the chemist from [1] work with the definition). Rubenstein et al. (2017) conducted pioneering work in establishing a formalism that discusses "exact transformations" between SCMs, which allowed the authors to (i) marginalize out 'irrelevant' variables, (ii) aggregate variables into sensible groups, and even (iii) view dynamic systems in a stationary way. Following that, Beckers and Halpern (2019) fixed several shortcomings by generalizing the former formalism to "(strong) abstractions" that (i) work on SCMs directly opposed to probabilistic parameterizations thereof and (ii) consider all possible interventions of an SCM opposed to only a selected subset.

We are given the standard formulation of an SCM $M = (\mathcal{U}, \mathcal{V}, \mathcal{R}, \mathcal{F}, \text{Pr})$. The elements of $M$ are exogenous variables $\mathcal{U}$ sometimes called 'unmodelled' terms that lie outside of what we are modelling, endogenous variables $\mathcal{V}$ that we actually model (e.g. DP in Fig.1), the range $\mathcal{R}$ of any of the variables, that is, the actual values they can take on (e.g. DP can be 'balanced'), the set of structural equations $\mathcal{F}$ that each dictate how each endogenous variable is caused through a given set of other variables in terms of an equation (e.g. HD = $f_i$(TC, $U_i$) where $f_i$ would be a mathematical description of the mechanism that makes TC affect HD) and finally Pr is a probability on the values of the exogenous terms, that is $\mathcal{R}(\mathcal{U})$, which implies a probability on the

endogenous terms as well. Since the discussed abstractions need be causal it is important for them to satisfy the causal prowess of the model pair under inspection, where model pair refers to some low- and high-level SCMs $(M_L, M_H)$ for which we want to either prove or falsify the hypothesis that $M_H$ is an abstraction of $M_L$. To achieve this, typically interventions are considered as the key element that needs to be aligned for the pair, therefore, we additionally equip our SCMs with a partially ordered set $\mathcal{I}$ that describes admissible interventions. For example $\mathcal{I} = \{\emptyset, \mathbf{V} \leftarrow \mathbf{v}\}$ would denote that only two types of interventions are admissible (i) the special case 'no intervention' denoted by the empty set or (ii) an intervention that sets all $\mathcal{V}$ variables simultaneously to some constant vector. With that, we can now give the state of the art formalization of a causal abstraction as:

**Definition 1** *If for some low- and high-level pair of SCMs denoted $(M_L, M_H)$ we have following conditions fulfilled for functions $\tau : \mathcal{R}_L(\mathcal{V}_L) \to \mathcal{R}_H(\mathcal{V}_H), \tau_{\mathcal{U}} : \mathcal{R}(\mathcal{U}_L) \to \mathcal{R}(\mathcal{U}_H), \omega_\tau : \mathcal{I}_L \to \mathcal{I}_H$:*

1. *all functions are surjective, $\omega_\tau$ further order-preserving*

2. *for all $i \in \mathcal{I}_L$ we have $\tau(\text{Pr}_L^i) = \text{Pr}_H^{\omega_\tau(i)}$*

3. *for all $(\mathbf{u}_L, i) \in \mathcal{U}_L \times \mathcal{I}_L$ we have $\tau(M_L(\mathbf{u}_L, i)) = M_H(\tau_{\mathcal{U}}(\mathbf{u}_L), \omega_\tau(i))$,*

*then $(\tau, \tau_{\mathcal{U}}, \omega_\tau)$ describe a $\tau$-abstraction from $M_L$ to $M_H$.*

Returning to the chemist example from [1], suitable formalizations for $M_L, M_H$ can be shown to commute via some $\tau$-abstraction for a corresponding choice of $\tau, \tau_{\mathcal{U}}$, and $\omega_\tau$. To

improve our intuition on the concepts introduced just now in Def.1, let us examine another toy example both formally and in more detail:

**Example 1** *The low-level SCM $M_L$ consists of $\mathcal{U}_L = \{U_1, U_2, U_3, U_4\} \subseteq \mathbb{R}^4, \mathcal{V} = \{A, B, C, D\}$ and*

$$\mathcal{F}_L = \{A = U_1, B = U_2, C = A+B+U_3, D = A+B+U_4\}$$

*whereas the high-level SCM $M_H$ is given by $\mathcal{U}_H = \{V_1, V_2\} \subseteq \mathbb{R}^2, \mathcal{V}_H = \{X, Y\}$ and*

$$\mathcal{F}_H = \{X = V_1, Y = 2 \cdot X + V_2\}.$$

*For the sake of simplicity, we will simply consider a deterministic setting ignoring* $\Pr$ *to prove our $\tau$-abstraction for $(M_L, M_H)$ more quickly. The admissible intervention sets are given by*

$$\mathcal{I}_L = \{\emptyset, (A, B) \leftarrow (a, b), (C, D) \leftarrow (c, d), (A..D) \leftarrow (a..d)\}$$

*and*

$$\mathcal{I}_H = \{\emptyset, X \leftarrow x, Y \leftarrow y, (X, Y) \leftarrow (x, y)\}.$$

*Then the function*

$$\tau : (A..D) \mapsto (A + B, C + D),$$

*with*

$$\tau_\mathcal{U} : (U_{1..4}) \mapsto (U_1 + U_2, U_3 + U_4)$$

*and*

$$\omega_\tau = \{(\emptyset \mapsto \emptyset), ([(A, B) \leftarrow (a, b)] \mapsto [X \leftarrow x]), \dots)$$

*form a $\tau$-abstraction from $M_L$ to $M_H$. It is easy to show that $\tau, \tau_\mathcal{U}, \omega_\tau$ are surjective and $\omega_\tau$ further order-preserving. For the sake of brevity we will skip the proofs, also for the equalities from Def.1, and instead show a simple example calculation*

$$
\begin{array}{ccc}
(U_{1..4}) = (2, 3, 1, 1) & \xrightarrow{\tau_\mathcal{U}} & (V_1, V_2) = (5, 2) \\
\downarrow M_L & & \downarrow M_H \\
(A..D) = (2, 3, 6, 6) & \xrightarrow{\tau} & (X, Y) = (5, 12)
\end{array}
$$

*illustrating the commutation.* ∎

**Learning.** To the best of our knowledge, there exist *no* works yet on actually learning $\tau$-abstractions i.e., no automation for neither the verification of whether some $(M_L, M_H)$ form an abstraction nor for actually acquiring some $\tau$ that would abstract $M_L$ to $M_H$. Knowing that the statement $S :=$"there exists a $\tau$ s.t. $M_H$ is a $\tau$-abstraction of $M_L$" is true would lend itself at most to an educated guess on what $\tau$ might look like. Typically, we don't know whether $S$ is true, nor do we know $M_L, M_H$.

**Related Work.** Javed, White, and Bengio (2020) investigated an online learning paradigm for detecting spurious features and thus learning better causal graphs. The two key differences here are (i) that there exists no mechanism for *abstracting* from one causal graph to another, however, the

final graph might very well align with the revised graph abstraction, and (ii) that abstractions operate on the SCM- and not just graph-level. Another work conducted by Chu, Rathbun, and Li (2020) investigated how observing new data points with each learning iteration can help with estimating better causal effects. One of the key challenges here being the decision function over previous representations and the aggregation procedure for revising previous representations to a best current estimate.

## [3] Future Work: Updating Existing Abstractions

While even just learning causal abstractions as in Def.1 remains an open problem, starting to work on *continual* causal abstractions poses a viable first step towards general causal abstraction learning.

**Step-by-Step Plan.** The following is a high-level description of an implementation of CCA in reference to Fig.1:

1. *Get the Initial Causal Abstraction.* Given our scientific question on the causal relationship of interest, that is whether the statement "$\exists f. Y = f(X)$" holds and if so what $f$ is, use the available data $\mathbf{D} \in [0, 1]^{m \times n}$ of $m$ rows and $n$ features (that include $X, Y$) to acquire a parameterized SCM $\hat{M}_L$. For instance, one could train a neural SCM (see Xia et al. (2021) for an introduction).

2. *Extract the Decision Rules.* For our binary setting, further assuming deterministic SCM for simplicity, simply take the predictions $\hat{P} := \hat{M}_L(\mathcal{R}(\mathcal{V}_L))$ over the ranges of the endogenous variables. Each input-output tuple will be a decision rule $r_i := (\mathcal{R}(\mathcal{V}_{L,i}), \hat{P}_i)$, read as '$r_{i,1} \implies r_{i,2}$', collectively forming a decision rule set $R$.

3. *Detect & Discard the Inconsistency.* Given a new data point $\mathbf{d}' \in [0, 1]^n$ check whether the decision rules predict the values of the data point, that is, whether $\forall(v, w) \in \mathbf{d}'. \exists r_i \in R. (v, w) = r_i$ holds. In our sumo practitioner example from Fig.1 we would observe that $(\text{TC} = 1, \text{HD} = 0) = r_{\text{TC}=1}$ since $r_{\text{TC}=1} = (1, 1)$. Remove (temporarily) all rules and data columns that covered the inconsistent predictor $j$ (here: $j = \text{TC}$), creating the alternate rule- and data-sets $R_{\setminus j}, \mathbf{D}_{\setminus j}$.

4. *Learn the $\tau$-Abstraction.* Optimize

$$\tau^* = \arg\max_\tau \mathcal{L}(\mathbf{D}_{\setminus j}, M_H(\tau))$$

for a suitable loss function $\mathcal{L}$ where $M_H(\tau)$ is a (neural) SCM with endogenous variables $\tau(\mathcal{V}_\mathcal{L})$.

**Key Shortcomings.** The laid out step-by-step plan lends itself to seemingly immediate implementation. However, the two major shortcomings are (i) that we do not know how to effectively search for $\tau$ (in our simple example we are at least able to resort to exhaustive search) and (ii) the assumption that the $n$ features from the initial data set will actually contain the features that end up in the final, continually updated abstraction (for the Fig.1 example this means that we actually observed LL/HL from the start). Another drawback is the assumption of extracting decision rules out of a learned model, which was granted in our case since we investigated a simple binary variable model.

**Acknowledgments** The authors thank the anonymous reviewers of the Bridge program for their valuable feedback. Furthermore, the authors acknowledge the support of the German Science Foundation (DFG) project "Causality, Argumentation, and Machine Learning" (CAML2, KE 1686/3-2) of the SPP 1999 "Robust Argumentation Machines" (RATIO). This work was supported by the Federal Ministry of Education and Research (BMBF; project "PlexPlain", FKZ 01IS19081). It benefited from the Hessian research priority programme LOEWE within the project WhiteBox, the HMWK cluster project "The Third Wave of AI" (3AI) & the National High-Performance Computing project for Computational Engineering Sciences (NHR4CES).

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
