# OpenReview forum: "Continual Causal Abstractions"
_AAAI.org/2023/Bridge/CCBridge — AAAI23 Bridge Continual Causality_

### Official Review · Reviewer_RrB9 · 2022-12-01
**Makes sense and fits the theme**

**Rating:** 6
**Confidence:** 4

**Review:**

This paper describes the problem of adjusting levels of abstraction for causal analysis in a continual fashion, as research questions change or more data become available. It reviews one of the prominent frameworks for causal abstraction and uses simple examples to illustrate the tasks and challengs. The problem makes sense and fits the theme of the program. A drawback of the paper is that only very preliminary and vague ideas toward a solution are indicated. I don't see a promising direction for addressing the problem in the present paper.

---

### Official Review · Reviewer_2RJY · 2022-12-05

**Rating:** 7
**Confidence:** 3

**Review:**

The paper looks at updating causal abstractions in a continual fashion. I think it is a good fit for the venue. I also found it well-written. I particularly liked the use of examples (especially in Figure 1) as a way of presenting some of the key ideas. This makes it much easier to read.

Definition 1 (and the line before it) introduces notation that is not defined. I know there is a lack of space, but I do not think there is any point having notation if it is not defined. This made Definition 1 very hard to understand.

---

### Official Review · Reviewer_cAVD · 2022-12-06
**Continual Causal Abstractions**

**Rating:** 8
**Confidence:** 4

**Review:**

This paper discusses how can we continually update the causal abstractions of the previous task for the newly arrived task in such a way so that the causal abstractions is consistent for both previous and current tasks. Authors suggest the exploration of  actual learning τ -abstractions i.e., automatic verification of abstractions for SCM. This is really an interesting direction to explore, although there are similar works in the intersection of causal inference and CL:

Javed, Khurram, Martha White, and Yoshua Bengio. "Learning causal models online." arXiv preprint arXiv:2006.07461 (2020).
Chu, Zhixuan, Stephen Rathbun, and Sheng Li. "Continual Lifelong Causal Effect Inference with Real World Evidence." (2020).

---

### Decision · Program_Chairs · 2022-12-05

**Decision:**

Accept

**Comment:**

Accept - Poster

The paper presents a short description of updating causal abstractions in a continual fashion. The topic is highly relevant to the bridge and all reviewers agreed that the contribution is interesting. Whereas all reviews were positive, some constructive suggestions for improvement have been made. This primarily includes a description of definition 1, which presently lacks an explanation and clarification of notation. In addition, it is suggested to provide some further references and make an attempt at presenting the idea in a more concrete way. We suggest that the authors include these suggestions in the extra page of the camera ready version.